# Use of Medicinal Mushrooms in Layer Ration

**DOI:** 10.3390/ani9121014

**Published:** 2019-11-21

**Authors:** Shad Mahfuz, Xiangshu Piao

**Affiliations:** State Key laboratory of Animal Nutrition, College of Animal Science and Technology, China Agricultural University, Beijing 100193, China; shadmahfuz@yahoo.com

**Keywords:** medicinal mushrooms, laying hens, health status, performance

## Abstract

**Simple Summary:**

The extensive use of antibiotics in the poultry industry to increase production performance has led to human health hazards. The use of natural herbs as antibiotic substitutes has been reported in the poultry feed industry. Therefore, the objective of this review was to determine the effect of different levels of mushrooms and their extract in diet on laying performance and health status. On the basis of previous findings, dietary supplementation using mushrooms as a natural feed supplement sustained laying performance and improved immunity in laying hens.

**Abstract:**

Application of different medicinal mushrooms intended to enhance production performance and health status has created an importance demand in poultry production. One goal of using medicinal mushrooms is to get rid of antibiotics in poultry feed without affecting the optimum performance. Increasing concerns about this issue have led to more attention on antibiotic substitutes and a significant demand for them for organic egg production. Thus, supplementation with medicinal mushrooms is a new concept for research in layer production, however, there is still a great deal of confusion about inclusion levels and the mode of action of medicinal mushrooms on production performance and health status in laying hens. Taking this into account, this review outlines the experimental uses of medicinal fungi on the growth performance, laying performance, egg quality, and health status of layer birds based on previous findings to date. Finally, we highlight that supplementation with medicinal fungi can play a role on the immunity, health, and production performance in laying hens.

## 1. Introduction

Traditionally, mushrooms have been used for highly valued food and pharmaceutical purposes because of their role as a tonic and their benefit to health [1]. Cultivated edible mushrooms are good sources of protein, have low-fat content, and are cholesterol free [2]. Mushrooms are also very popular as a quality protein containing essential amino acids, adequate vitamins, minerals, and are rich source of different unsaturated fatty acids [3]. Different bioactive components have been extracted from the fruiting body and mycelium part of mushroom and tested in invitro studies. Polysaccharides are considered to be the most activate component in mushrooms which have immune stimulating activities [3]. In addition, the polysaccharides in mushroom have been found to produce different cytokines and increase the weight of immune stimulating organs in laboratory animals [4,5]. Presently, researchers have become interested in the role of medicinal mushrooms in poultry production systems.

Antibiotics as feed additives have been used as growth and health promoters in poultry production [6], however, because of the appearance of microorganisms that are resistant to specific antibiotics, the application of antibiotics in poultry ration has been forbidden or restricted in the developed countries [7,8]. As a result, exploring new growth-promoting alternatives to antibiotics has become a hot topic of research for several years [9]. Chickens are very sensitive to immunosuppressive stressors and infectious diseases [10]. Different infections are responsible for reduction in growth rates, poor egg production, and mortality, which have resulted in huge economic losses in the poultry industry. There is a direct relationship between feeding and the immune system of the host [11]. Various attempts, through genetic manipulation, dietary alterations, various medicinal supplements, etc. have been tried to reduce the cholesterol content in meat and eggs, and therefore improve their health status [12].

At present, there are several scientific works about the health promoting benefits of involving mushrooms in farm animals. Currently, poultry researchers are committed to using unconventional natural feed supplement as a substitute for antibiotics that have been proven as possible ways to enhance the health and to improve the production in poultry. Although it is known that mushrooms are medicinally importance for chickens health, unfortunately, the inclusion level of mushrooms in poultry diets is still under consideration. Findings from past reports have highlighted that their inclusion may enhance production performance and health in laying hens [13]. Taking this into consideration, this review is focused on the importance of the medicinal mushrooms as an alternative for antibiotics that can improve the performance and the immunity in laying hens.

## 2. Common Medicinal Mushrooms Used in Layer Study

A group of mushrooms have been identified as medicinal mushrooms, in recent years, due to their biological properties both invivo and invitro studies. The phylum Basidiomycota is the most predominant among the mushrooms species that has been proven to be a medicinal mushroom [14]. On the basis of some previous studies, we have identified some common medicinal mushrooms that can be used as a source of active substances for optimum performance and health status in layer chickens. A list of major medicinal mushrooms that were used in poultry ration during the previous years is presented in Table 1 and Figure 1.

## 3. Biological Role of Medicinal Mushrooms

Mushrooms have been reported to have many useful functions including antitumor, anticancer, antihypertensive, cholesterol lowering effect, antioxidant properties, anti-inflammatory, immune-modulatory function, as well as anti-bacterial, antiviral, and antifungal activities on human and animal health [15,16].

### 3.1. Antitumor Activities

The shiitake mushroom (*Lentinus edodes*) is rich in antitumor agents which play a role inhibiting cancer cell growth [17]. Aqueous extracts from the vegetative submerged mycelia of cultivated *Ganoderma lucidu*, and *Lentinus edodes* have been reported to have antitumor activities [18]. *F. velutipes* mushrooms have been reported to hold bioactive compound having antitumor functions [19]. The extract of *F. velutipes* mushroom has been used to oppose breast cancer cells [20]. Recently, significant novel components with anticancer function were discovered in *F. velutipes* by Chinese researchers. These researchers discovered a sesquiterpene, which is known as flammulinol A, along with other flammulinolides A–G derived from *F. velutipes* mushroom that were effective against several cancer cell lines [21]. A recent study by Dong et al. [22] reported that polysaccharide, purified from *Ganoderma applanatum* mushroom, was effective against human breast cancer in an invitro study.

### 3.2. Antioxidant Activities

Today, the antioxidant properties of different medicinal mushrooms are well-known. Some previous studies have reported that the polysaccharides and oligosaccharide present in medicinal mushrooms show antioxidant functions [23]. Conventional uses of butylated hydroxyanisole and butylated hydroxytoluene as synthetic antioxidants can be hazardous to humans, and therefore there is a need to discover natural antioxidant products [24]. Tang et al. [3] stated that the phenolic ingredients present in mushrooms may have the capacity to withdraw the oxidation of the LDL for their anti-inflammatory activities. A fibrinolytic enzyme that was successfully purified and derived from the culture supernatant of needle mushroom was reported by Park et al. [25]. The antioxidant activities depend on different parts and varieties of mushrooms. Zeng et al. [26] stated that *F. velutipes* mushroom hold a higher phenolic amount with the highest antioxidant activities. Different mushrooms were found to exhibit vitaminC and selenium that can play a role in antioxidant functions [14]. A recent study by Lin et al. [27] found that the *Cordyceps sobolifera* (Ascomycetes) mushroom exhibits antioxidant properties as a functional food and dietary supplement. In addition, *Agaricus brasiliensis* are considered potential auxiliaries for the treatment of patients with rheumatoid arthritis due to their capacity to reduce oxidative stress [28]. The anti-inflammatory and antioxidant properties of *A. bisporus* biomass extracts from an in vitro culture were reported by Muszynska et al. [29]. In their studies, incubation of Caco-2 cells with *A. bisporus* extracts resulted in decreased expression of cyclooxygenase-2 and prostaglandin F2α receptor as compared with the lipopolysaccharide (LPS) or TNF-α-activated cells. The antioxidant activity of *Pleurotus ostreatoroseus* (Agaricomycetes) mushroom was also noted by Brugnari et al. [16].

### 3.3. Lipid Metabolism Activities

The positive role of golden needle mushroom on lipid metabolism in male hamsters was reported by Yeh et al. [30]. Their study showed that both the extract and the powder originating from needle mushroom were capable of reducing serum and liver tissue cholesterol level in hamsters. Another study by Yang et al. [31] found a lower level of plasma triglyceride, total cholesterol (TC), and low-density lipoprotein cholesterol in diet-induced hyperlipidemic rats fed *Hericiumerinaceus* mushroom exo-polymer. Lovastatin, as well as γ-aminobutyric acid (GABA), were identified from *F. velutipes* fruiting bodies [32]. Lovastatin is used to reduce cholesterol production that can diminish risks of heart diseases [32,33]. Another study by Harada et al. [34] reported very effective results by decreasing the systolic pressure in rats using GABA-mediated *F. velutipes* mushroom powder. *ß*-D-glucan and its derivatives present in medicinal mushrooms ensured their cholesterol lowering effects by reducing the absorption or increasing the faecal excretion [35]. The oyster mushroom is also famous for its cholesterol reducing functions [36].

### 3.4. Antimicrobial Activities

The antimicrobial properties of medicinal mushrooms are well established. The extracts derived from medicinal *Pleurotus* species mushroom have been reported to have potential antibacterial and antifungal functions [37,38]. An invitro experiment was conducted by Sknepnek et al. [39] with reishi mushroom (*Ganoderma lucidum*) on antimicrobial functions. Their studies concluded that the liquid *Ganoderma lucidum* mushroom beverage at a 0.04 mg/mL concentration was very useful against *Staphylococcus epidermidis* and *Rhodococcus equi*. In addition, it was very useful against *Bacillus spizizenii, B. cereus,* and *R. equi* at a 0.16 mg/mL concentration. Nedelkoska et al. [40] reported that the mushroom fruiting body was very effective against different bacteria. Kashina et al. [41] stated that the mushroom, *F. velutipes,* exhibited inhibitory activities in opposition to two different harmful fungi (*Sporothrix schenckii* and *Candida albicans*). Enokipodins have been found in the needle mushroom that has antimicrobial functions [42].

### 3.5. Immune Functions

The immune functions of mushrooms are well known. Different protein and various peptides present in mushrooms are able to modify the immune response positively [14]. Invitro immune-modulatory studies with *F velutipes* showed that raw 264.7 cells were stimulated to secret nitric oxide upon administration of 200 to 500 µg/mL *F.velutipes* polysaccharide (FVPA2). The FVPA2 also encouraged the proliferation of the spleen lymphocytes and B lymphocytes in experimental mice [43]. Manayi et al. [44] used the extract of *Ganoderma applanatum* mushroom at a concentration of 1000 mg/kg diet on the defense mechanisms in rainbow trout. This study found the potential ability of *G. applanatum* mushroom extract to activate immunologic parameters in rainbow trout. Lee et al. [45] found that the mushroom could increase the concentration of IFNγ that has a toxic function against lymphoma cell. The polysaccharides of needle mushroom were found to produce different cytokines and increase the weight of immune stimulating organs in laboratory animals [4,5]. The mushroom polysaccharides increased the body weight of experimental mice and the weight ratio of the thymus and spleen, as well as it could modulate the T cell subpopulation of thymocytes and splenocytes [30]. Moreover, the polysaccharides of mushroom increased NO (nitric oxide), TNF-a, IL-1b, and IL-6 production, and lymphocyte proliferation in mice model [46].

### 3.6. Nutritional Roles

Mushrooms are very popular for their nutritional values. Mushrooms have been reported as a good source of six major nutrients which include carbohydrates, especially dietary fiber, proteins, vitamins, minerals, lipids, and water. Rich in proteins, carbohydrates, and fiber with low fat are the unique features of the medicinal mushroom. In addition, different types of essential amino acids (AA) have been found in mushroom [47,48,49]. The nutritional component of different mushroom showed dry matter (DM) 74% to 89.6%; crude protein (CP) 8.9% to 14.8%; carbohydrate 43.33% to 69.40%; total detergent fiber (TDF) 1.9% to 7.40%; crude fat (EE) 1.75% to 3.91%; ash (total mineral) 4.91% to 8.40%; calcium (Ca) 2.21% to 3.05%, and phosphorus (P) 1.68% to 1.88% [3,50,51]. 

## 4. Medicinal Mushrooms in Layer Chicken Ration

The data regarding the role of medicinal mushrooms in layer chicken performance are summarized in Table 2. Hence, there has been considerable debate regarding current findings on laying hens’ performance, as well as many variables that have been associated with the current findings such as mushroom species, use dosage, method of application (either non-fermented or fermented with beneficial organisms), part of the mushroom (either fruiting bodies or stem base), and the treatment period. However, collectively, many scientists agreed that mushrooms could have a positive role by improving the laying percent, table egg quality, egg yolk cholesterol level, as well as immunity in laying hens. Further studies are needed to detect the actual dose for optimum performance in layer chickens. 

### 4.1. Application of Medicinal Mushrooms on Performance and Egg Quality

There have been limited studies conducted, in previous years, to evaluate the effects of medicinal mushrooms in laying hens. Mahfuz et al. [52] conducted a study to examine the role of *Fammulina velutipes* mushroom stem wastes (FVW) on growth performance, and immunity in pullet birds on basic of different levels (2%, 4%, and 6%). They found that the final live weight was greater (*p* < 0.05) in mushroom fed groups at all levels (2%, 4%, and 6%) than that of the control and antibiotics diets. No differences (*p* > 0.05) were found for the average daily feed intake, average daily weight gain, and feed conversion ratio (FCR) among treatments. Dry matter (DM), crude protein (CP), and ether extract (EE) retention were higher (*p* < 0.05) in FVW diets than the control and antibiotic diets. Excreta DM was higher (*p* < 0.05) and pH was lower (*p* < 0.05) in FVW diets than the control and antibiotic diets. The higher body weight, in this study, must be related to higher nutrient retention in mushroom supplemented groups. In addition, the Excreta DM was higher in the mushroom supplemented groups which suggests that incorporated FVW reduced excreta moisture, which can prevent the wet litter in poultry house, as well as increase the absorption of nutrients and reduce the ammonia gas production from excreta in chicken house. Teye et al. [58] stated that high moisture content, high temperatures, and high pH can facilitate the production of ammonia from excreta. Mahfuz et al. [53], consequently, assessed the role of *Flammulina velutipes* in laying hens ration with different levels of mushrooms (2%, 4%, and 6%) in experimental diets and did not find any differences (*p* > 0.05) in laying performance parameters such as average daily egg production percentage, egg mass, FCR, etc., in laying hens. The number of unmarketable table eggs was fewer (*p* < 0.05) in mushroom fed diets as compared with the control diets. This study also found suitability for calcium retention in eggshells with FVW diets as compared with the control and the antibiotic diets. It was hypothesized that the higher calcium retention could be related to a higher number of marketable eggs in mushroom fed groups. No effects on egg production percentage, egg mass, and FCR ensured the fact that feeding mushroom did not have any adverse effects on laying performance. Lee et al. [13] found that feeding *F. velutipes* mycelium had no adverse effects on egg production percentage, feed intake, and FCR, in laying hens, but the egg weight was found greater (*p* < 0.05) in the 1% and 3% mushroom feeding groups than the control diets. Furthermore, feeding mushroom at the 4% level resulted in significantly higher (*p* < 0.05) egg albumen height, haugh unit, eggshell weight, and shell thickness. It was thought that mushrooms contain higher level of CP that leads to increased egg albumin and might have an effect on shell gland for continuous eggshell formation. On the contrary, Na et al. [59] found that dietary inclusion of mushroom had no effect on eggshell weight, shell thickness, and haugh unit. Finally, the authors suggested that mushroom, as a natural resource of feed for laying hens, can be used at the 5% level without affecting normal performance.

*Lentinula edodes* are commonly known as Shiitake mushroom which has long been considered to be a medicinal mushroom. An experiment was conducted with the shiitake mushroom on laying performance and egg quality by Hwang et al. [54]. Higher (*p* < 0.05) egg production and higher (*p* < 0.05) haugh unit in eggs were reported by feeding shiitake mushroom than that of the control group. However, the other laying parameters including egg weight, shape index, shell thickness, albumen height, yolk color, and the egg sensory evaluation (e.g., appearance, color, flavor, oily nature) were not affected (*p*> 0.05). Egg yolk fatty acids, especially linoleic acid, total omega-6 fatty acid (n-6), and the polyunsaturated fatty acid were found to be higher (*p* < 0.05) in 0.5% mushroom feeding groups than the control group, however, palmitoleic acid and α-linolenic acid were lower (*p* < 0.05) in the 0.5% mushroom feeding group than the control fed group. In addition, the cholesterol concentration of egg yolk was lower (*p* < 0.05) in the 0.5% mushroom fed diet than the control group [54]. Foods rich in total omega-3 fatty acid (n-3), total omega-6 fatty acid (n-6), as well as polyunsaturated fatty acid (PUFA) are very helpful for health. A diet enriched withn-3 PUFA is considered to have preventative functions for people with vascular diseases [60]. Willis et al. [61] reported that adding *Lentinula edodes* mushroom mycelium extract to layer diets had no significant effects on laying performance. In a subsequent study, Willis et al. [62] further investigated that birds fed with fungus myceliated grain could successfully induce molting and allowed egg production earlier than the control group. This study concluded that fungus myceliated meal can be an effective alternative to conventional feed withdrawal methods, for the successful initiation of molt and maintenance of post-molt performance.

The mushroom, *Cordyceps militaris,* was used in layer diets to evaluate the performance and egg yolk cholesterol level by Wang et al. [57]. The results showed significantly lower (*p* < 0.05) egg cholesterol in mushroom fed groups with 1% and 2% levels than the control fed groups. In addition, improved (*p* < 0.05) FCR with greater (*p* < 0.05) egg weight were found at the 2% level mushroom fed group than in the control group. However, no significant differences were observed on the eggshell weight, egg yolk weight, shell thickness, and egg yolk color, among the treatment groups.

Lee et al. [13] used the spent mushroom (*Hypsizygus marmoreus*) substrates in layer ration to evaluate the feeding effects on egg production and table egg quality. None of the production performance parameters were affected by feeding mushroom *Hypsizygus marmoreus* during the entire study period. However, the egg yolk color scores were higher (*p* < 0.05) in mushroom fed groups than the control group. They concluded that fermented spent mushroom can be used, up to15% in layer ration, without affecting normal laying percent and table egg quality.

A wild medicinal mushroom, *Ganoderma lucidum,* was used for pullet performance in a study by Ogbe et al. [55]. No significant effects were observed for the feed intake by feeding *Ganoderma lucidum* mushroom in pullets. However, FCR was improved (*p* < 0.05) in mushroom groups than the non supplemented control group. Lee et al. [56] used *Pleurotus eryngii* mushroom to evaluate the performance in laying hens. Egg cholesterol level was lower (*p* < 0.05) in the mushroom groups than the control diets, however they did not observe any significant differences on egg production performance by feeding mushroom base diets in laying hens. The haugh unit was greater (*p* < 0.05) in the 1% and 2% experimental diet groups. The authors finally concluded that laying hens fed with the residue of *Pleurotus eryngii* mushroom could produce lower cholesterol in eggs. 

### 4.2. Application of Medicinal Mushrooms on Health Status in Layer Chickens 

Body immunity and inner organ weight are good indicator of health status in chickens. No significant difference was found for inner relative organ (liver, gizzard, spleen, and abdominal fat) weights between control and antibiotic fed diets [52], however, the bursa weight was higher (*p* < 0.05) in the mushroom fed diets than the control and antibiotic fed diets [52]. No effects on inner organ weights ensured that feeding mushroom did not have any toxic effects on pullet chickens, whereas the higher bursa weight ensured the improved immune status in experimental chickens. Higher bursa weight is an indicator of better health status and a sound physiological response to body immune system [63]. The appropriate level of immune sub-parameters such as immunoglobulin, cytokines, protein, and some biochemical index are important to maintaining the immune response in host. Antibody titers against Newcastle disease (ND), infectious bronchitis (IB), and Avian influenza (AI) virus vaccines were found to be higher (*p* < 0.05) in mushroom stem waste fed diets in pullet [52]. In addition, serum immunoglobulin parameters (IgA, IgG, and IgM) were found to be higher (*p* < 0.05) in mushroom fed diets than the control and antibiotic fed diets in the experimental pullet. Bai et al. [64] stated that serum immunoglobulin concentrations can generate humoral immune response in animals due to their important roles on immune function fighting against various infections. In addition, supplementation of β-glucan from edible mushroom had a significant immune stimulatory effect in chickens [65]. Similarly, the antibody titers against infectious bursal diseases virus were greater in mushroom *Ganoderma lucidum* fed groups than the control diets, in pullet [55]. In another study by Mahfuz et al. [66] antibody response on ND was greater (*p* < 0.05) in the 6% mushroom stem fed group and IB were greater (*p* < 0.05) in all levels of mushroom fed groups than both the positive and negative control diets, in laying hens. This study further demonstrated that the serum cytokines concentrations (IL-2, IL-6, IL-4, and TNF-α) were higher (*p* < 0.05) in mushroom feed groups than the control and antibiotic fed groups, in laying hens. The polysaccharides in medicinal mushrooms have strong immune modulatory activity and possess antioxidant activity that could enhance nonspecific and specific immune responses invitro [46]. In addition, cytokines are known to be regulators of the immune status. The function of IL-2 relies on the commencement of B and T lymphocytes cells. However, the activation of Th1 depends on secretion of IL-2, TNF-α, along with other cytokines that create the cellular immunity [67]. Similarly, Jarosz et al. [68] reported that the function of Th2 depends on IL-4, with other cytokines secretion that stimulates humoral immunity.

Lee et al. [13] found that the number of pathogenic bacteria, especially *Salmonella* spp., *E coli,* and *Clostridium* spp. were lower (*p* < 0.05) in mushroom (*Flammulina velutipes*) fed groups than the control diets, in laying hens. Similarly, Willis et al. [61] reported that adding *Lentinula edodes* mushroom mycelium extract in layer diets, could decrease the number of pathogenic bacteria *Salmonella* spp. in the caecum and crop of birds fed with mushroom extracts. Lee et al. [56] used mushroom *Pleurotus eryngii* in layer ration. This study found that both the serum triglyceride and the serum cholesterol were lowered (*p* < 0.05) in mushroom fed groups than the control. Moreover, the dietary inclusion of dried mushroom at the 1% and 2% level showed greater (*p* < 0.05) serum antioxidant enzyme activities, in laying hens. This is due to a higher content of phenolic substance and different minerals, especially selenium, in mushrooms. Dietary supplementing selenium could enhance the body weight gain and antioxidant enzyme activities in chickens [69]. The author finally concluded that the residue of *Pleurotus eryngii* mushroom could improve the antioxidant status in layer chickens. Sun et al. [70] reported that edible mushrooms have a hypo-cholesterolemic effect on health and suggested its use as an oral medicine. The improved antioxidant status of chickens fed with different medicinal mushrooms was due to the presence of phenolic compounds, especially phenolic acid, which is the major naturally occurring antioxidant components found in mushrooms.

## 5. Conclusions and Future Perspectives

This review highlights that medicinal mushrooms could be fruitfully used as an effective natural growth promoter, as well as an immune boosting agent, in layer birds. In spite of the brood uses of medicinal mushrooms in layer diets, further studies by various researchers are recommended regarding the dosages of medicinal mushrooms on optimum performance and immune response, in laying hens. Therefore, future study should examine the use of medicinal mushroom in reaction to a pathogen challenge, as well as dosages. We suggest future research on medicinal mushrooms as alternates for antibiotics in laying hens so that it can be an effective strategy for organic egg production and encourage future researchers to discover the aspects of medicinal mushrooms that are important to immunity and health status that previous studies were not able to explore.

## Figures and Tables

**Figure 1 animals-09-01014-f001:**
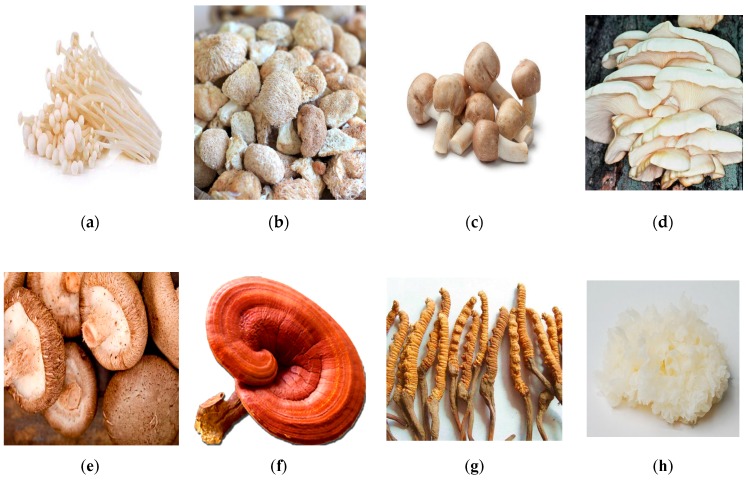
Photographs of different medicinal mushrooms: (**a**) Flammulina velutipes, (**b**) *Hericium erinaceus*, (**c**) Agaricus brasiliensis, (**d**) Pleurotus ostreatus, (**e**) *Lentinula edodes*, (**f**) Ganoderma lucidum, (**g**) *Cordyceps sinensis*, and (**h**) Tremella fuciformis.

**Table 1 animals-09-01014-t001:** Botanical classification and distribution of medicinal mushrooms used in poultry ration ^1^.

Common Name/Local Name	Scientific Classification	Distribution
Golden needle mushroom/Winter mushroom/Lily mushroom/Velvent shank/Enoki mushroom/Jingen Gu	K: Fungi P: Basidiomycota C: Agaricomycetes O: Agaricales F: Physalacriaceae G: Flammulina Sp: *Flammulina velutipes*	Europe, USA, and Asia, especially China, Japan, Korea, and Vietnam
Monkey’s head/Lion’s mane/Bear’s head/Yamabushitake (Japan)/Houtou or Shishigashira (China)	K: Fungi P: Basidiomycota C: Agaricomycetes O: Russulales F: Hericiaceae G: *Hericium* Sp: *Hericium erinaceus/Hericium caput-medusae*	Europe, Asia, and North America
White button mushroom/Almond mushroom/ Mushroom of sun/God’s mushroom	K: Fungi P: Basidiomycota C: Agaricomycetes O: Agaricales F: Agaricaceae G: Agaricus Sp: *Agaricus brasiliensis/Agaricus bisporus*	California, Hawaii, Great Britain, The Netherlands, Taiwan, Philippines, Australia, Brazil, China, Japan, Korea and Vietnam
Oyster mushroom	K: Fungi P: Basidiomycota C: Agaricomycetes O: Agaricales F: Pleurotaceae G: Pleurotus Sp: *Pleurotus ostreatus/Pleurotus eryngii*	All over the world, especially Germany, India, China, Japan, and Korea
Shiitake mushroom	K: Fungi P: Basidiomycota C: Agaricomycetes O: Agaricales F: Marasmiaceae G: Lentinula Sp: *Lentinulaedodes*	Southeast Asia, especially China and Japan
Reishi/Lingzhi mushroom	K: Fungi P: Basidiomycota C: Agaricomycetes O: Polyporales F: Ganodermataceae G: Ganoderma Sp: *Ganoderma lucidum/Ganoderma applanatum,*	Southeast Asia especially China, Japan, and Korea
Caterpillar Mushroom/Cordyceps mushroom	K: Fungi P: Ascomycota C: Sordariomycetes O: Hypocreales F: Cordycipitaceae G: *Cordyceps* Sp: *Cordyceps sinensis/Cordyceps militaris*	Asian countries, e.g., Nepal, China, Japan, Bhutan, Korea, Vietnam, and Thailand
Snow fungus/ Snow ear/Silver ear fungus/White jelly mushroom.	K: Fungi P: Basidiomycota C: Tremellomycetes O: Tremellales F: Tremellaceae G: Tremella Sp: *Tremella fuciformis*	North America, Africa, Australia, New Zealand, Asia including Korea, Japan, and China

^1^ K, kingdom; P, phylum; C, class; O, order; F, family; G, genus; and Sp, species.

**Table 2 animals-09-01014-t002:** Role of medicinal mushrooms on performance in layer chickens ^1^.

Mushroom Species	Study Design	Main Finding	References
*Flammulina velutipes*	ISA Brown layer pullet from 10 weeks to 16 weeks (42 days) form: dried mushroom, dose: mushroom 2%, 4%, 6% (inclusion type)	●increased final live weight ●increased nutrient retention ●higher dry matter content in excreta ●lower pH in excreta ●higher bursa weight ●higher antibody titers against ND, IB, and AI ●higher serum immunoglobulin IgA, IgG, and IgM	Mahfuz et al. [52]
*Flammulina velutipes*	ISA Brown layer from 19 weeks to 29 weeks (70 days) form: dried mushroom, dose: mushroom 2%, 4%, 6% (inclusion type)	●increased marketable egg number ●increased calcium retention ●higher antibody titers against ND and IB ●higher serum immunoglobulin sIgA, IgG, and IgM ●higher serum cytokines IL-2, IL-4, IL-6, and TNF-α	Mahfuz et al. [53]
*Flammulina velutipes*	Hy-line Brown layer from 60 weeks to 65 weeks (35 days) form: mushroom fermented by *Bacillus subtilis* and *Klebsiella spp*, dose: mushroom 1%, 2%, 3%, 4%, 5% (supplementation type)	●higher egg weight ●higher albumen height, haugh unit, eggshell weight, and shell thickness ● lower cecal *Salmonella spp* and *E. coli* number ● lower excreta ammonia (NH_3_) concentration	Lee et al. [13]
*Lentinula edodes*	Tetran Brown layer from 22 weeks to 30 weeks (56 days) form: dried mushroom powder, dose: mushroom 0.25%, 0.5% (supplementation type)	●higher egg production ●higher haugh unit ● higher linoleic acid, total n-6 and polyunsaturated fatty acid in egg yolk ● lower egg yolk cholesterol	Hwang et al. [54]
*Ganoderma lucidum*	Lorman Brown pullet from 0 to 20 weeks form: dried mushroom powder dose: mushroom 2 g/kg, 1 g/kg, 0.5 g/kg (supplementation type)	●improve FCR ●higher antibody titers	Ogbe et al. [55]
*Pleurotus eryngii*	Hendrix layer from 22weeks to 30 weeks (56 days) form: dried mushroom powder dose: mushroom 0.5%, 1%, 2% (supplementation type)	●lower egg yolk and serum cholesterol ●higher haugh unit ●higher antioxidant enzyme activities	Lee et al. [56]
*Cordyceps militaris*	Hendrix layer from 22 weeks to 34 weeks (84 days), Form: dried mushroom waste dose: mushroom 5 g/kg, 10 g/kg, 20 g/kg, (supplementation type)	●lower egg yolk cholesterol ●higher egg mass ●improved FCR	Wang et al. [57]

^1^ ND, Newcastle disease; IB, infectious bronchitis (IB); AI, Avian influenza; Ig, immunoglobulin; IL, interleukin; n-6, omega-6 fatty acid; and FCR, feed conversion ratio.

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
