# Peer review of "Use of Medicinal Mushrooms in Layer Ration"

_animals, 2019, doi:10.3390/ani9121014_

Round 1
Reviewer 1 Report
L79 to L166 are devoted to human
Table 2 is not well organized
pullets must be separated from Laying hens.
Use the same units in table 2 (% or g/kg) and use the same units for the same experiment (g/kg, table 2, ref 52) and % in L184.
table 2 is not usefull since the same data are presented from L180 to L294
Author Response
Dear Sir
Good day. Thank you very much for your kind consideration with our submitted article and offering us the further opportunity to submit the revised manuscript. Please find here the point to point comments with necessary changes as per suggested with this file below. We have revised our manuscript for language and grammar checked by a native English speaker working in our University. We do thanks to skilled reviewers, academic editors and editorial board members as well for their critical evaluation to make the manuscript more effective for review process in Animals Journal.
L79 to L166 are devoted to human
Responses: Dear Professor, Thank you very much for your comments. Actually, in this part (L 79-166) we have highlighted the medicinal values of different mushrooms on in-vitro and some laboratory animals, to explain the importance’s of mushrooms in farm animal study. This information may help animal researchers to do further research with mushroom and their byproducts in farm animals as well as chickens.
Table 2 is not well organized, pullets must be separated from Laying hens. Use the same units in table 2 (% or g/kg) and use the same units for the same experiment (g/kg, table 2, ref 52) and % in L184.
Responses: Thank you. Table 2 was revised as per suggestion. We have designed the information about pullet first and then we have put the information about laying hens both in Table and Text. We have used the unit in percent (%) in table and text with the same experiment. Please check the red color marking points in Table 2.
table 2 is not usefull since the same data are presented from L180 to L294
Responses: Thank you very much. Actually we have summarized only the significant findings in tabular form (table 2). And in the text, we have presented all the past experiment with mushrooms in laying hens with details findings from the published literature. We hope the future readers and interested researchers can easily find out the previous significant findings with the experimental design in tabular form.
Thank you.
Many thanks.
Sincerely yours,
Prof. Dr. Xiang Shu Piao,
State Key laboratory of Animal Nutrition, College of Animal Science and Technology, China Agricultural University, Beijing 100193, China
Corresponding Author,
Email: piaoxsh@cau.edu.cn; Tel./Fax.: +86-1062733688
Reviewer 2 Report
Antibiotics supplementation as a feed additive in poultry and other farm animals has been ban in some countries around the globe. Hence the search for alternatives to antibiotics usage has attracted increasing scientific studies lately.
ln this review paper, "Uses of medicinal mushrooms in layer ration", the authors tried to discuss previous works and hence giving in depth knowledge on the impact of adding different concentrations of medicinal mushrooms in the diet of the layer hen. l found the work of Dr. Mahfuz and Dr. Piao rich and would add a lot of information to the already existing scientific knowledge when it undergoes a minor revision. Therefore, below are the comments to which the others should revise:
The grammatical language of the entire manuscript should be improved. example: ln line 17: .........the health status has created an importance demand.........; line 21 delete, "about".......and about....; line 25 delete, "to" ....findings till to date.; line 26 change sound health to health; line 44 delete "the", As a results,.....; line 51 change Till to date, to Till date. and many more throughout the manuscript. ln line 24, change "layer" to layers or layer birds. as in: ......health status in layer ln line 41,:.... growth promoters and health status.... can be written.as.. growth and health promoter in poultry..... ln Table 1. under Distribution, kindly delete "over the" ln line 82. rewrite line 111-line114. please discuss the major findings of the works (Muszynska et al., and Brugnari et.al.). line 119 to 121. rewrite line 132 to 1350. ls reishi mushroom the same as G. lucidum mushroom? if yes creat the link in the text. line 195. delete "conducted another experiment to" line 223. space "with n-3" line 240. space "to 15% line 254 to 259" The bursa...................... Please remove this section, rewrite it and past in line 299. Before "lmproved antioxidant....。" line 261. change ..... are important on maintains...。 to...... are important in maintaining......... line 296. Better, Conclusion and further perspectives. 1. rewrite the conclusion, highlighting the key findings of this study. 2. suggest further research direction or requirements base on the findings. 3. correct grammatical errors. References. 1. the abbreviations for some Journal names are wrong. example, Frontiers microbiol. is Front Microbiol., Annals of Anim. Sci. is wrong. please cross check. 2. some journal names are not abbreviated while others are. please kindly cross check with the Journals referencing style. example. line 495.Author Response
Dear Sir
Good day. Thank you very much for your kind consideration with our submitted article and offering us the further opportunity to submit the revised manuscript. Please find here the point to point comments with necessary changes as per suggested with this file below. We have revised our manuscript for language and grammar checked by a native English speaker working in our University. We do thanks to skilled reviewers, academic editors and editorial board members as well for their critical evaluation to make the manuscript more effective for review process in Animals Journal.
Antibiotics supplementation as a feed additive in poultry and other farm animals has been ban in some countries around the globe. Hence the search for alternatives to antibiotics usage has attracted increasing scientific studies lately.
ln this review paper, "Uses of medicinal mushrooms in layer ration", the authors tried to discuss previous works and hence giving in depth knowledge on the impact of adding different concentrations of medicinal mushrooms in the diet of the layer hen. l found the work of Dr. Mahfuz and Dr. Piao rich and would add a lot of information to the already existing scientific knowledge when it undergoes a minor revision. Therefore, below are the comments to which the others should revise:
The grammatical language of the entire manuscript should be improved. example: ln line 17: .........the health status has created an importance demand.........; line 21 delete, "about".......and about....; line 25 delete, "to" ....findings till to date.; line 26 change sound health to health; line 44 delete "the", As a results,.....; line 51 change Till to date, to Till date. and many more throughout the manuscript. ln line 24, change "layer" to layers or layer birds. as in: ......health status in layer ln line 41,:.... growth promoters and health status.... can be written.as.. growth and health promoter in poultry..... ln Table 1. under Distribution, kindly delete "over the" ln line 82. rewrite line 111-line114. please discuss the major findings of the works (Muszynska et al., and Brugnari et.al.). line 119 to 121. rewrite line 132 to 1350. ls reishi mushroom the same as G. lucidum mushroom? if yes creat the link in the text. line 195. delete "conducted another experiment to" line 223. space "with n-3" line 240. space "to 15% line 254 to 259" The bursa...................... Please remove this section, rewrite it and past in line 299. Before "lmproved antioxidant....。" line 261. change ..... are important on maintains...。 to...... are important in maintaining......... line 296. Better, Conclusion and further perspectives. 1. rewrite the conclusion, highlighting the key findings of this study. 2. suggest further research direction or requirements base on the findings. 3. correct grammatical errors. References. 1. the abbreviations for some Journal names are wrong. example, Frontiers microbiol. is Front Microbiol., Annals of Anim. Sci. is wrong. please cross check. 2. some journal names are not abbreviated while others are. please kindly cross check with the Journals referencing style. example. line 495.
Responses: Dear Professor, Thank you very much for your comments and suggestion. As per your advice, the manuscript was checked by a native speaker working in our university for English grammar and language. And we have acknowledged the English teacher name in Acknowledge portion.
In Text: As per your kind advises we have thoroughly revised our manuscript and did the necessary changes as per suggestion; please check the red color highlighted sentences in the revised file and Table. Please check the line number: L-22,25,26-27; 43,46,54,92-93, 118-122;128-130, 142-145,209,240,273-276,281, 310.Thank you.
In Conclusion: we have revised it as per advises. Please check the highlighted sentences with L-320-321,324-328, Thank you.
In References: We have rechecked the references format according to Journal style. Please check the highlighted sentences with L-345, 486, 490, 498, 528, 535. Thank you.
Many thanks.
Sincerely yours,
Prof. Dr. Xiang Shu Piao,
State Key laboratory of Animal Nutrition, College of Animal Science and Technology, China Agricultural University, Beijing 100193, China
Corresponding Author,
Email: piaoxsh@cau.edu.cn; Tel./Fax.: +86-1062733688
Round 2
Reviewer 1 Report
The language and style must be improved.
Some examples:
-L9 and L10, L36-37:.....mycelium "and tested" in in vitro studies, L 38: which have, L56: ways instead of therapies, L90: which have ... L140: have been, L273: are instead of is
-L43-44: long, many years=too much,
Table 2 is not usefull
some hypotheses are surprising and not supported, for example L269-270.
Author Response
Revision Note-R2 (List of modification) Date: 2019-11-13 (y-m-d)
Manuscript ID: animals-631400- R2.
Dear Sir
Good day. Thank you very much for your kind consideration with our submitted article and offering us the further opportunity to submit the revised manuscript. Please find here the point to point expert reviewer’s comments with necessary changes as per suggested with this attached file. We have revised our manuscript for language and grammar checked by a native English speaker working in our University. We do thanks to skilled reviewers, academic editors and editorial board members as well for their critical evaluation to make the manuscript more effective for review process in Animals Journal.
Many thanks.
Sincerely yours,
Prof. Dr. Xiang Shu Piao,
State Key laboratory of Animal Nutrition, College of Animal Science and Technology, China Agricultural University, Beijing 100193, China
Corresponding Author,
Email: piaoxsh@cau.edu.cn; Tel./Fax.: +86-1062733688
Comments: (Reviewer-1)-R2
-The language and style must be improved.
Responses: Dear Professor, thank you very much. We have further revised the MS for language. The necessary changes have been made with the following lines (red color); Please check the line number: L- 12;21;26;35;41;50;58-60; 65;136;142;155;169-170;179;1801-81;184;194;199;202; 206-207; 208; 211-212;216;226;262; 291; 301;303;306;308-309; 310;313-314.
-L9 and L10, L36-37:.....mycelium "and tested" in in vitro studies, L 38: which have, L56: ways instead of therapies, L90: which have ... L140: have been, L273: are instead of is; L43-44: long, many years=too much,
Responses: Dear Professor, Thank you very much for your good comments. We have rewritten and revised the suggested part as per advises (red color), Please check the line number 9-10; 36-38; 43-44;56; 90;140; 270;
-some hypotheses are surprising and not supported, for example L269-270.
Responses: Thank you. We have deleted the contradictory statement from the text as per advises.
-Table 2 is not usefull
Responses: Thank you very much. In table 2, we have summarized only the significant findings in tabular form. And in the text, we have presented the details findings from the published literature. We hope, the tabular form will be more helpful to the future researchers to set up a new experiment. Thus we like to put the Table 2 in the text. We hope your kind consideration.
Thank you very much.